# Is There a Risk for Semaglutide Misuse? Focus on the Food and Drug Administration’s FDA Adverse Events Reporting System (FAERS) Pharmacovigilance Dataset

**DOI:** 10.3390/ph16070994

**Published:** 2023-07-11

**Authors:** Stefania Chiappini, Rachel Vickers-Smith, Daniel Harris, G. Duccio Papanti Pelletier, John Martin Corkery, Amira Guirguis, Giovanni Martinotti, Stefano L. Sensi, Fabrizio Schifano

**Affiliations:** 1UniCamillus University, Via di S. Alessandro 8, 00131 Rome, Italy; 2Psychopharmacology, Drug Misuse and Novel Psychoactive Substances Research Unit, School of Life and Medical Sciences, University of Hertfordshire, Hatfield AL10 9AB, UK; ducciopapanti@gmail.com (G.D.P.P.); j.corkery@herts.ac.uk (J.M.C.); giovanni.martinotti@gmail.com (G.M.); f.schifano@herts.ac.uk (F.S.); 3Department of Epidemiology and Environmental Health, University of Kentucky College of Public Health, 111 Washington Avenue, Lexington, KY 40536, USA; rachel.vickers@uky.edu; 4Institute for Pharmaceutical Outcomes and Policy, University of Kentucky College of Pharmacy, 289 South Limestone Street, Lexington, KY 40536, USA; daniel.harris@uky.edu; 5Center for Clinical and Translational Sciences, University of Kentucky, 800 Rose Street, Lexington, KY 40506, USA; 6Cividale Community Mental Health Centre, ASUFC Mental Health Department, Via Carraria 29, 33043 Cividale del Friuli, Italy; 7Department of Pharmacy, Swansea University Medical School, Swansea SA2 8PP, UK; amira.guirguis@swansea.ac.uk; 8Department of Neurosciences, Imaging, and Clinical Sciences, University of Chieti-Pescara, 66100 Chieti, Italy; stefano.sensi@unich.it; 9Center for Advanced Studies and Technology (CAST), Institute of Advanced Biomedical Technology (ITAB), University of Chieti-Pescara, Via dei Vestini 21, 66100 Chieti, Italy

**Keywords:** semaglutide, drug misuse, drug abuse, pharmacovigilance, image- and performance-enhancing drugs (IPEDs), glucagon-like peptide-1 (GLP-1) agonists

## Abstract

Recent media reports commented about a possible issue of the misuse of antidiabetics related to molecules promoted as a weight-loss treatment in non-obese people. We evaluated here available pharmacovigilance misuse/abuse signals related to *semaglutide*, a glucagon-like peptide-1 (GLP-1) analogue, in comparison to other GLP-1 receptor agonists (*albiglutide*, *dulaglutide*, *exenatide*, *liraglutide*, *lixisenatide*, and *tirzepatide*) and the *phentermine*–*topiramate* combination. To acheieve that aim, we analyzed the Food and Drug Administration’s FDA Adverse Events Reporting System (FAERS) dataset, performing a descriptive analysis of adverse event reports (AERs) and calculating related pharmacovigilance measures, including the reporting odds ratio (ROR) and the proportional reporting ratio (PRR). During January 2018–December 2022, a total of 31,542 AERs involving the selected molecules were submitted to FAERS; most involved dulaglutide (n = 11,858; 37.6%) and semaglutide (n = 8249; 26.1%). In comparing semaglutide vs. the remaining molecules, the respective PRR values of the AERs ‘drug abuse’, ‘drug withdrawal syndrome’, ‘prescription drug used without a prescription’, and ‘intentional product use issue’ were 4.05, 4.05, 3.60, and 1.80 (all < 0.01). The same comparisons of semaglutide vs. the phentermine–topiramate combination were not associated with any significant differences. To the best of our knowledge, this is the first study documenting the misuse/abuse potential of semaglutide in comparison with other GLP1 analogues and the phentermine–topiramate combination. The current findings will need to be confirmed by further empirical investigations to fully understand the safety profile of those molecules.

## 1. Introduction

Type 2 diabetes mellitus (T2DM) is the most common form of diabetes and is a chronic and progressive illness [1]. In parallel with this, the worldwide prevalence of obesity, a key target in the treatment and prevention of diabetes [2], has been progressively increasing over the past few decades and is predicted to continue to rise in coming years. However, lifestyle modification, including dietary changes and physical exercise, is often insufficient to achieve clinically meaningful weight loss due to physiological mechanisms that limit weight reduction and promote weight regain [3].

In the absence of contraindications, metformin has traditionally been considered as the first-line medication in T2DM patients; however, it promotes modest weight reduction [4,5]. Conversely, the invasive bariatric surgery has been the primary approach to treat severely obese individuals. More recently, the emergence of agents impacting on the brain satiety centers’ function may provide effective, non-invasive treatment of obesity for individuals with and without diabetes. The long-acting glucagon-like peptide-1 receptor agonist (GLP-1 RA) semaglutide, and tirzepatide, a dual-agonist at both the receptors for glucagon-like peptide-1 (GLP-1) and the glucose-dependent insulinotropic polypeptide (GIP), are now approaching the success seen with bariatric surgery; in association with these molecules, decreasing body weight (e.g., by >10% in most patients), with a favorable safety profile, is being observed [2,6,7,8]. Indeed, tirzepatide showed even better dose-dependent efficacy (e.g., greater reduction in hemoglobin A1c/HbA1c and body weight) than placebo, basal insulin, and the two GLP-1 analogues dulaglutide and semaglutide [6,9,10,11,12,13,14]. The principal role of the incretins GIP and GLP-1 has generally been thought to stimulate insulin secretion. GLP-1 improves fasting blood glucose due to its direct action on pancreatic islets and decreases postprandial hyperglycemia due to inhibition of gastric emptying, and thus reducing levels of glucose entry into the circulation. GIP directly stimulates insulin secretion through the β cell GIP receptors. Indirectly, GIP potentiates α cell activity to enhance α to β cell communication through the GLP-1R/glucagon receptor (GcgR), thus indirectly stimulating insulin secretion through the α cell [15]. Among other incretin mimetics, both liraglutide, at a maximum daily dose of 3.0 mg, and semaglutide, at a maximum weekly maintenance dose of 2.4 mg, have already received regulatory approval for the treatment of obesity, whilst tirzepatide is currently being assessed for this indication [10,16]. In the United Kingdom (UK), semaglutide received approval as a weight-loss medication to be prescribed on the National Health System (NHS) as of March 2023 [17]. With these medications, a range of gastrointestinal adverse effects have been reported, e.g., nausea and diarrhea [18]; conversely, severe hypoglycemia, fatal adverse events, acute pancreatitis, cholelithiasis, and cholecystitis are considered rare events [1,8].

### Is There a GLP-1 Analogue Misuse Issue?

Classical medications for obesity management can be typically divided into a few groups: opiate antagonists (naltrexone); pro-dopaminergic drugs (bupropion, phendimetrazine, benzphetamine, diethylpropion, and phentermine), fat blockers (orlistat), and the antihyperglycaemic drug metformin; and the serotonin 2C receptor agonist lorcaserin, the serotonin and norepinephrine reuptake inhibitor sibutramine, and the selective cannabinoid receptor-1 blocker rimonabant, which was lately dismissed from the market for severe human health risks, e.g., psychiatric side effects, especially depression and suicide [19]. Because of their misuse potential, some of these molecules are classified as controlled substances, e.g., lorcaserin and sibutramine are both classified as Schedule IV controlled drugs under the Controlled Substances Act [19,20]. Similarly, because of phentermine’s relationship with amphetamine, it was determined to have the potential for abuse and designated as a Schedule IV controlled substance [21]. Thus, used in combination with topiramate, both synergism and efficacy should be improved, with side effects and potential misuse being possibly reduced [22]. In the case of metformin, it has been associated with levels of both diversion [23] and abuse [24], having been ingested either at high dosages [25] or by eating-disorder individuals [26].

A large range of newspaper [27] reports commented about the possible existence of the issue of misuse of semaglutide and other GLP-1 analogues. In line with this, in March 2023 the French National Agency for Drug Safety announced ‘enhanced surveillance’ levels for semaglutide. In fact, since September 2022, the drug agency had been alerted by both a range of videos on social media and by pharmacists reporting forged prescriptions and use for weight loss in non-diabetics [28]. Social media platforms’ semaglutide promotion as a weight-loss treatment [29], and the associated increase in demand, may well have contributed to an ongoing worldwide shortage of the drug [29,30,31]. One could hypothesize that the non-realistic versions of physical attractiveness being promoted for otherwise healthy, non-obese people may be behind these putative levels of misuse. The issue can be facilitated by the putative acquisition of medications from rogue websites [32,33].

In contrast with the above, it is of interest that, by the time of the drafting of this paper, no literature reports focusing on the issues of misuse of semaglutide and novel antidiabetics appeared to have been published. Hence, we aimed here to determine the available pharmacovigilance misuse/abuse signals relating to *semaglutide* versus other incretin mimetics, such as the molecules *albiglutide*, *dulaglutide*, *exenatide*, *liraglutide*, *lixisenatide*, and *tirzepatide*, and the combination *phentermine–topiramate*, by analyzing the Food and Drug Administration’s FDA Adverse Events Reporting System (FAERS) dataset.

## 2. Results

From January 2018 to December 2022, a total of 31,542 adverse event reports (AERs) involving the selected molecules were submitted to FAERS. Among these, 37.6% involved dulaglutide (n = 11,858), 26.1% semaglutide (n = 8249), 25.0% liraglutide (n = 7883), 8.2% exenatide (n = 2585), 1.24% lixisenatide (n = 390), 0.9% tirzepatide (n = 290), 0.6% phentermine–topiramate (n = 183), and 0.3% albiglutide (n = 104). Regarding semaglutide, during the selected timeframe, an increase in the number of reported AERs compared to remaining molecules was observed (Figure 1); overall, most reports came from the United States of America (USA) and involved female adults (Table 1).

Most of the AERs recorded for semaglutide were related to gastrointestinal issues; an off-label medication use was recorded here only for semaglutide (483/8249 cases; 5.85%) (Table 2).

In terms of reported outcomes by drug, semaglutide was associated with fatalities in 273/8249 (3.3%) of AERs, whilst this event occurred with the remaining molecules in 1705/23,110 (7.4%) of AERs (Table 3).

### Pharmacovigilance Signals

Drug misuse-, abuse-, and withdrawal-related AERs were most typically reported for semaglutide compared with the other selected GLP-1 analogues (dulaglutide, liraglutide, exenatide, lixisenatide, tirzepatide, and albiglutide) and the combination phentermine–topiramate (Table 3). Specifically, ‘drug abuse’, ‘drug withdrawal syndrome’, and ‘prescription drug used without a prescription’ were reported >3.50 times as frequently (e.g., PRR values were 4.05, 4.05, and 3.60, respectively; FDR < 0.01), and ‘intentional product use issue’ was reported almost two times as frequently (PRR = 1.80; FDR < 0.01) (Table 4). Conversely, no significant differences in terms of the selected AER occurrence were identified when comparing semaglutide vs. the phentermine–topiramate combination.

## 3. Discussion

To the best of our knowledge, this is the first study documenting the misuse and abuse potential of semaglutide in comparison with both remaining GLP-1 analogues (dulaglutide, liraglutide, exenatide, lixisenatide, tirzepatide, and albiglutide) and the phentermine–topiramate combination. The comparison was carried out here with the help of a range of worldwide, valuable [34,35], pharmacovigilance data, namely those derived from the FAERS. AERs related to semaglutide showed steady and progressive increasing levels during the years 2018–2022; conversely, the phentermine–topiramate-related AERs remained roughly stable, and AERs related to the remaining GLP-1 mimetics appeared to have decreased during the same years. Consistent with this, in the years previous to the timeframe considered here (i.e., 2018–2022), prescriptions for all the molecules selected increased, and this was especially true for those relating to liraglutide, dulaglutide, and semaglutide [36,37,38,39].

As expected, for both semaglutide and the other GLP-1 analogues, most reported AERs involved gastrointestinal adverse events [40]. Conversely, the phentermine–topiramate AERs most typically involved dizziness, headache, blurred vision, hypoesthesia, and paresthesia, which have all been reported for topiramate [41]. In relation to the AERs registered for the phentermine/topiramate combination, it is interesting that no significant data on misuse/abuse are reported, despite clinicians’ concerns regarding the use of single phentermine [21,22]. Given both the low risk of semaglutide severe adverse events (e.g., mostly mild-to-moderate and transient), and its beneficial metabolic and cardiovascular actions [42,43,44], the molecule is considered to possess an overall favorable risk/benefit profile for patients with T2DM [43]. It may be a reason of concern, however, that an off-label prescription issue for semaglutide, but not for the remaining GLP-1 analogues, was identified here. This may happen when a drug is being used for an unapproved indication/population or at an unapproved dosage. Off-label use of a medication is at times associated with its misuse potential. In line with this, and in contrast with remaining GLP-1-RA analogues, semaglutide appeared here to be associated with significantly higher levels of (i) abuse, (ii) intentional product use issues, and (iii) use without a prescription. To the best of our knowledge, this finding has never been reported before in the medical literature and is fully consistent with the vast range of anecdotal, unconfirmed magazine and newspaper reports [17,27,30,31].

### 3.1. Semaglutide and GLP-1 RA as Image- and Performance-Enhancing Drugs (IPEDs)

The phenomenon of drug misuse and abuse for weight-loss purposes has been frequently reported in the literature. Image- and performance-enhancing drugs (IPEDs) include a wide range of drugs across various pharmacological categories, which are misused to obtain an alteration/enhancement of physical performance or appearance [45], as occurs with slimming products. A vast range of molecules have been misused as weight-loss agents, especially relating to sympathomimetic agents, e.g., amphetamine/methamphetamine-type drugs, ecstasy, and cocaine [46]. Other extreme slimming misusing agents have included β-2 agonists such as clenbuterol [47], diuretics [48], and dinitrophenol/DNP [49]. Overall, these agents are being misused and abused by vulnerable, non-obese, body-dysmorphic subjects for their image-enhancing [50] and significant slimming potential. From this point of view, semaglutide may possess the potential to be misused as a weight-loss IPED agent. Phentermine is one of the medications being considered by obesity specialists for weight-loss purposes [19]. Whilst phentermine has not been previously associated with evidence of physical dependence or addiction [51], it may well possess the potential for misuse [52]. This may explain the lack of statistical differences, in terms of misuse-/abuse-related AERs, when the phentermine–topiramate combination was compared here with semaglutide. Overall, the information provided by this study does not allow us to deduce explanations for the misuse of each individual substance but, on the basis of pharmacovigilance signals, enables us to compare molecules with each other and provide useful information for clinicians and institutions to monitor possible side effects, adverse events, and misuse. We can hypothesize that issues related to the formulation (subcutaneous versus oral), drug availability and ease of prescription, pharmacokinetic and pharmacodynamic properties, and the effects on weight reduction (semaglutide and liraglutide would appear to be the most effective in the long term) of the individual drugs may explain the misuse of semaglutide in comparison with the other molecules under study.

### 3.2. Semaglutide and GLP-1 RAs as Molecules Acting on the Reward System?

One could wonder if there is a neurobiological issue putatively associated with the decreased appetite and improved levels of satiety [2,53], and hence with the significant weight loss, associated with GLP-1 RA in general and semaglutide in particular. In the central nervous system (CNS), GLP-1Rs are expressed in several brain regions involved in the regulation of metabolism and energy balance [54]. The reward-related brain regions that regulate appetitive and consumption behaviours include the mesolimbic regions’ nucleus accumbens (NAc), ventral tegmental area (VTA), and amygdala [55,56], with GLP-1Rs being located in the mesolimbic system [57,58]. Furthermore, the gut–brain axis peptide ghrelin enhances dopamine release in the VTA [59]. Hence, one could wonder if the withdrawal symptoms associated here with semaglutide may be conceivably related with its effect on the reward system; this is consistent with a range of anecdotal Redditors’ observations highlighting both rebound and craving phenomena after withdrawing from semaglutide [60]. In line with this, a recent randomized-controlled trial showed that 1 year after having withdrawn from a weekly treatment of 2.4 mg of semaglutide, participants regained two-thirds of their weight loss [61]. Indeed, GLP-1 analogues may reduce the rewarding effects of palatable food intake (for a review, see also [59]) and the food-related brain responses in both T2DM and obese subjects in insula, amygdala, putamen, and orbitofrontal cortex [62]. Consistent with these observations, GLP-1 RA can reduce cue- and drug-induced fentanyl seeking [63]; physical and behavioral effects of morphine withdrawal [58]; cue-, stress-, and drug-induced heroin seeking [64]; and use of cocaine, amphetamine, alcohol, and nicotine in animals when administered over several days or weeks [64,65,66,67]. In one randomized-controlled trial, exenatide, administered weekly in combination with nicotine replacement therapy, improved smoking abstinence, reduced craving and withdrawal symptoms, and decreased weight gain among abstainers [68]. Another study demonstrated that weekly exenatide significantly reduced heavy drinking days and total alcohol intake in a subgroup of obese patients [69]. Therefore, GLP-1 RA is gaining increasing attention as a new therapeutic agent for the treatment of reward-system-related disorders [57].

### 3.3. The Potential Use of GLP-1 RA in Neurology

Preclinical studies suggest that exenatide can normalize dopaminergic function, showing its potential protective action against cytokine-mediating apoptosis and its substantial therapeutic utility in diseases where neuroinflammation may play a significant role, such as Parkinson’s disease (PD) [70], Alzheimer disease (AD), diabetes, and strokes [71]. Consistently, exenatide administered weekly had positive effects on practically defined off-medication motor scores in PD, and this outcome was sustained beyond the period of exposure [72]. Interestingly, the GLP-1 RA exenatide, in its extended-release formulation, has been successfully employed to alleviate the motor symptoms of PD patients [73], a condition characterized by a deficit in dopaminergic neurotransmission, a critical axis of the reward system. Notably, in preclinical models of brain aging and neurodegeneration, the same compound has been shown to positively affect the signaling of the brain-derived neurotrophic factor (BDNF) and modulate synaptic plasticity [71,74], thereby introducing the possibility of long-lasting behavioral effects driven by long-term structural brain and neural modifications.

### 3.4. Limitations

Despite the interest of the current findings, they may need to be interpreted with caution. First, although disproportionality analysis is a suitable tool for quantifying signs of drug abuse/misuse, it has a limited ability to differentiate the type or reason for abuse (e.g., recreational and self-medication). Furthermore, confounding factors such as comorbidities, dosages/routes of administration, and concomitant drugs consumed cannot be adequately assessed with a pharmacovigilance approach. This is due to the inherent nature of the reports used as primary sources for the study, reflecting only the information provided to the FDA by the reporter. The study of AERs alone is rarely sufficient to confirm that a certain effect in a patient was caused by a specific drug, as it may also have been caused by the disease treated, by a new disease developed by the patient, or by another medicine the patient is taking. Indeed, the number of case reports for a particular drug or suspected adverse reaction does not only depend on the actual frequency of the adverse reaction, but also on the extent and conditions of use of the drug, the nature of the reaction, the related public awareness levels, and adherence to reporting. Notwithstanding the related limitations and biasing factors of pharmacovigilance studies based on spontaneous reporting, the PRR and remaining statistical values identified here should be interpreted as strong signals of disproportionality.

## 4. Materials and Methods

### 4.1. Data Source

The present study focused on the FAERS pharmacovigilance data in relation to semaglutide and other drugs within the same clinical indication and in the same drug group, including albiglutide, dulaglutide, exenatide, liraglutide, lixisenatide, and tirzepatide. Although not in the same drug class, the phentermine–topiramate combination, being used with the same indication (e.g., obesity), was included here. FAERS data were made available online through the FAERS Public Dashboard [75]. In pharmacovigilance, ‘misuse’ is considered the intentional and inappropriate use of a product other than as prescribed or not in accordance with the authorized product information, whilst ‘abuse’ is the intentional non-therapeutic use of a product for a perceived reward, including ‘getting high’/euphoria (for an overview, see [76]). Since the focus here was on misuse, abuse, and diversion issues, the preferred terms (PTs) for the present analysis were selected from the standardized Medical Dictionary for Regulatory Activities (MedDRA) Query (SMQ) ‘Drug abuse, dependence and withdrawal’ [77]; these included: ‘Drug abuse’, ‘Substance abuse’, ‘Intentional product misuse’, ‘Dependence’, ‘Drug withdrawal syndrome’, ‘Withdrawal’, and ‘Withdrawal syndrome’. PTs possibly suggesting an abuse event (e.g., ‘Intentional product use issue’, ‘Overdose’, and ‘Prescription drug used without a prescription’) were also examined here.

### 4.2. Data Analysis

A descriptive analysis of the characteristics of AE reports, including sociodemographic data, country of origin, and concomitant licit/illicit substances, was performed here. IBM SPSS Statistics for Windows, version 28 (IBM Corp., Armonk, NY, USA) was used for all descriptive analyses. Pharmacovigilance reporting measures, including reporting odds ratio (ROR), proportional reporting ratio (PRR), information component (IC), and Bayesian empirical geometric mean (EBGM), were calculated for each dataset, using the R package PhViD [78]. All four pharmacovigilance measures were computed due to differences in their sensitivity and early detection potential to assess disproportionality, i.e., whether a drug–AE pair occurs more often than expected as defined by the false discovery rate (FDR), based on a 2 × 2 contingency table (Table 5) [79,80]. To identify signals above thresholds, the use of FDR, with an FDR < 0.05 to indicate significance, was used here [81,82].

Calculations for the frequentist measures, *ROR* [83] and *PRR* [84], are as follows:$$ROR=\frac{a\times d}{b\times c}$$
where the *ROR* describes the odds of the AE of interest among specific drug reports compared to the odds of the AE of interest among all other drug reports; a large *ROR* suggests that the drug–AE pair is reported more often, i.e., disproportionately, than expected.
$$PRR=\frac{a/(a+b)}{c/(c+d)}$$
which can be interpreted as the proportion of reports with the AE of interest among specific drug reports compared to the proportion of reports with the AE of interest among all other drugs; a large *PRR* suggests that the drug–AE pair is reported more often than expected.

The IC [85] and EBGM [86] are Bayesian measures with more complex computations, as described in detail elsewhere. If the lower endpoint of the 95% credible interval for the IC (i.e., IC025) is greater than zero (i.e., positive) then it suggests that the drug–AE pair occurs in the data more often than expected. Similarly, if the lower bound of the 90% credible interval of the EBGM (i.e., EB05) is large, this suggests that the drug–AE pair occurs in the data disproportionately.

## 5. Conclusions

The current findings, which indicate a possible semaglutide misuse/abuse issue, will need to be confirmed by further empirical investigations. These will help in better elucidating the GLP-1R agonists’ central pharmacodynamics, e.g., their interaction with a different range of receptors, the levels of availability of GLP1-R agonists for acquisition from rogue websites, and, with the help of properly designed epidemiological studies, the characteristics of their possible misuse/abuse in both the general and vulnerable populations.

## Figures and Tables

**Figure 1 pharmaceuticals-16-00994-f001:**
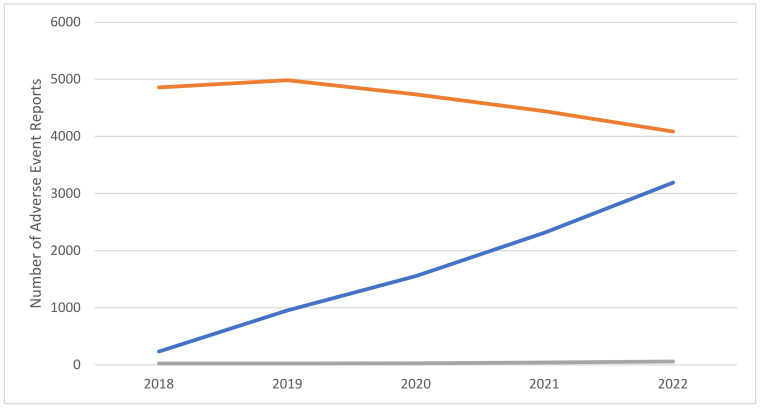
Number of adverse event reports (AERs) involving semaglutide, other glucagon-like peptide-1 (GLP-1) analogues (dulaglutide, liraglutide, exenatide, lixisenatide, tirzepatide, and albiglutide), and the combination phentermine–topiramate. Data source: Food and Drug Administration (FDA) Adverse Event Reporting System (FAERS; 2018–2022). Legend: semaglutide: blue color; other GLP-1 analogues: orange; phentermine–topiramate: grey.

**Table 1 pharmaceuticals-16-00994-t001:** Demographics related to adverse events (AEs) typically reported for semaglutide, phentermine–topiramate, and other glucagon-like peptide-1 (GLP-1) analogues (dulaglutide, liraglutide, exenatide, lixisenatide, tirzepatide, and albiglutide). Data source: Food and Drug Administration (FDA) Adverse Event Reporting System (FAERS; 2018–2022).

Number of AERs (%)	Overall	Semaglutide	Phentermine–Topiramate	Other GLP-1 Analogues *
**Mean Age, years (SD)**	61.0 (19.2)	60.2 (13.7)	49.9 (14.7)	61.4 (20.8)
**Females**	16,559 (53%)	4470 (54%)	156 (85%)	11,933 (52%)
**Males**	12,986 (41%)	3449 (42%)	22 (12%)	9515 (41%)
**Concomitant substances (%)**				
Alcohol	23 (0.1%)	2 (0.0%)	0 (0.0%)	21 (0.0%)
Cannabis	33 (0.1%)	13 (0.2%)	0 (0.0%)	20 (0.0%)
Cocaine	0 (0%)	0 (0.0%)	0 (0.0%)	0 (0.0%)
Opioids	1712 (5.4%)	249 (3.0%)	16 (8.7%)	1447 (8.7%)
Amphetamines	25 (0.1%)	9 (0.1%)	1 (0.0%)	16 (0.1%)
Benzodiazepines	1550 (4.9%)	238 (2.9%)	17 (9.3%)	1295 (5.6%)
**Country of origin**	USA 19,664 (62.0%)	USA 5016 (71.0%)	USA 173 (95%)	USA 14,475 (62.0%)
	France 1729 (6.0%)	Canada 825 (10.0%)	Korea 9 (5.0%)	France 1449 (6.0%)
	Canada 1562 (5.0%)	United Kingdom 360 (4.0%)	Not specified (0.0%)	Japan 1078 (5.0%)

*Abbreviations:* AER: adverse event report; GLP-1-RA: glucagon-like peptide-1 receptor agonists; SD: standard deviation; USA: United States of America. * This combines albiglutide, dulaglutide, exenatide, liraglutide, lixisenatide, and tirzepatide.

**Table 2 pharmaceuticals-16-00994-t002:** Most typically reported semaglutide, phentermine–topiramate, and other glucagon-like peptide-1 (GLP-1) analogues (dulaglutide, liraglutide, exenatide, lixisenatide, tirzepatide, and albiglutide) adverse event reports (AERs). Data source: Food and Drug Administration (FDA) Adverse Event Reporting System (FAERS; 2018–2022).

Semaglutide		Phentermine–Topiramate		Other GLP-1 Analogues *	
Preferred Term	# AER (%)	Preferred Term	# AER (%)	Preferred Term	# AER (%)
Nausea	1047 (13%)	Nephrolithiasis	14 (8%)	Nausea	1843 (8%)
Vomiting	921 (11%)	Headache	11 (6%)	Blood glucose increased	1604 (7%)
Diarrhea	699 (8%)	Weight increased	10 (5%)	Vomiting	1586 (7%)
Pancreatitis	492 (6%)	Angle closure glaucoma	9 (5%)	Pancreatitis	1459 (6%)
Off-label use	483 (6%)	Blurred vision	9 (5%)	Diarrhea	1426 (6%)
Weight decreased	465 (6%)	Suicidal ideation	8 (4%)	Acute kidney injury	1112 (5%)
Blood glucose increased	424 (5%)	Chronic kidney disease	7 (4%)	Weight decrease	1082 (5%)
Decreased appetite	387 (5%)	Hypoesthesia	7 (4%)	Fatigue	794 (3%)
Fatigue	357 (4%)	Breast cancer	6 (3%)	Decreased appetite	711 (3%)
Dehydration	352 (4%)	Paresthesia	6 (3%)	Chronic kidney disease	689 (3%)

Note that multiple preferred terms can be listed in an adverse event report. *Abbreviations:* AER: adverse event report; GLP-1: glucagon-like peptide. * This combines albiglutide, dulaglutide, exenatide, liraglutide, lixisenatide, and tirzepatide.

**Table 3 pharmaceuticals-16-00994-t003:** Most typically reported semaglutide, phentermine–topiramate, and other glucagon-like peptide-1 (GLP-1) analogues (dulaglutide, liraglutide, exenatide, lixisenatide, tirzepatide, and albiglutide). Data source: Food and Drug Administration (FDA) Adverse Event Reporting System (FAERS; 2018–2022).

Semaglutide		Phentermine–Topiramate		Other GLP-1 Analogues *	
Outcome	# AER (%)	Outcome	# AER (%)	Outcome	# AER (%)
Other outcomes	5418 (66%)	Other outcomes	154 (84%)	Other outcomes	14,206 (61%)
Hospitalized	3479 (42%)	Hospitalized	46 (25%)	Hospitalized	10,287 (45%)
Life threatening	306 (4%)	Disabled	14 (8%)	**Died**	**1705 (7%)**
Disabled	299 (4%)	Life threatening	3 (2%)	Life threatening	1103 (5%)
**Died**	**273 (3%)**	**Died**	**1 (1%)**	Disabled	671 (3%)
Required intervention	67 (1%)	Required intervention	1 (1%)	Required intervention	76 (<1%)

Note that multiple outcomes can be listed in an adverse event report. *Abbreviations:* AER: adverse event report; GLP-1: glucagon-like peptide-1. * This combines albiglutide, dulaglutide, exenatide, liraglutide, lixisenatide, and tirzepatide.

**Table 4 pharmaceuticals-16-00994-t004:** Signal scores regarding drug misuse-, abuse-, and withdrawal-related AERs for semaglutide, other glucagon-like peptide-1 (GLP-1) analogues (dulaglutide, liraglutide, exenatide, lixisenatide, tirzepatide, and albiglutide), and the phentermine–topiramate combination. Data source: Food and Drug Administration (FDA) Adverse Event Reporting System (FAERS; 2018–2022).

	Semaglutide vs. other GLP-1 Analogues				Semaglutide vs. Phentermine–Topiramate			
**PT (MedDRA)**	PRR (FDR)	ROR (FDR)	IC025 (FDR)	EB05 (FDR)	PRR (FDR)	ROR (FDR)	IC025 (FDR)	EB05 (FDR)
**Accidental overdose**	0.59 (0.60)	0.59 (0.60)	−1.62 (0.34)	0.50 (0.41)	Inf (<0.01)	Inf (<0.01)	−1.41 (0.50)	0.99 (0.52)
**Drug abuse**	**4.05 (<0.01)**	**4.05 (<0.01)**	−0.63 (0.16)	0.80 (0.12)	Inf (<0.01)	Inf (<0.01)	−1.74 (0.52)	0.99 (0.53)
**Drug level increased**	0.85 (0.46)	0.85 (0.46)	−1.12 (0.27)	0.62 (0.29)	Inf (<0.01)	Inf (<0.01)	−1.21 (0.49)	0.99 (0.52)
**Drug withdrawal syndrome**	**4.05 (<0.01)**	**4.05 (<0.01)**	−0.63 (0.16)	0.80 (0.12)	Inf (<0.01)	Inf (<0.01)	−1.74 (0.52)	0.99 (0.53)
**Incorrect route of product administration**	0.55 (0.61)	0.55 (0.61)	−1.65 (0.34)	0.48 (0.42)	Inf (<0.01)	Inf (<0.01)	−1.34 (0.50)	0.99 (0.52)
**Intentional product misuse**	0.42 (0.64)	0.42 (0.64)	−1.68 (0.35)	0.40 (0.45)	0.32 (<0.01)	0.32 (<0.01)	−1.01 (0.48)	0.99 (0.53)
**Intentional product use issue**	**1.80 (<0.01)**	**1.80 (<0.01)**	**0.08 (<0.01)**	**1.11 (<0.01)**	Inf (<0.01)	Inf (<0.01)	−0.54 (0.41)	0.99 (0.50)
**Overdose**	0.92 (0.46)	0.92 (0.46)	−0.66 (0.17)	0.72 (0.19)	Inf (<0.01)	Inf (<0.01)	−0.71 (0.44)	0.99 (0.51)
**Prescription drug used without a prescription**	**3.60 (<0.01)**	**3.60 (<0.01)**	−0.42 (0.10)	0.85 (0.08)	Inf (<0.01)	Inf (<0.01)	−1.50 (0.51)	0.99 (0.53)
**Substance use**	Inf (0.70)	Inf (0.70)	−0.29 (0.06)	**0.91 (0.04)**	Inf (0.04)	Inf (0.04)	−1.74 (0.53)	0.99 (0.53)

Boldface denotes significance at FDR < 0.05; Signal scores for drug–event pairs less than 5 are not shown. *Abbreviations:* EB05: Bayesian empirical geometric mean (lower 5th percentile of the posterior observed-to expected distribution); FDR: false discovery rate; GLP-1: glucagon-like peptide-1; IC025: information component; the IC025 value is the lower limit of a 95% credibility interval for the IC; MedDRA: Medical Dictionary for Regulatory Activities; PRR = proportional reporting ratio; PT: preferred terms; ROR: reporting odds ratio.

**Table 5 pharmaceuticals-16-00994-t005:** Disproportionality computation of drug–adverse event (AE) pairs.

	# Reports with AE of Interest	# Reports without AE of Interest
# Reports with drug of interest	A	b
# Reports without drug of interest	C	d

## Data Availability

The FDA Adverse Event Reporting System data are publicly available and can be found here: https://www.fda.gov/drugs/questions-and-answers-fdas-adverse-event-reporting-system-faers/fda-adverse-event-reporting-system-faers-public-dashboard (accessed on 4 July 2023).

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
