# Peer review of "Is There a Risk for Semaglutide Misuse? Focus on the Food and Drug Administration’s FDA Adverse Events Reporting System (FAERS) Pharmacovigilance Dataset"

_pharmaceuticals, 2023, doi:10.3390/ph16070994_

Round 1

Reviewer 1 Report

Since the Authors write about incretin drugs, some additional information should be provided regarding their mechanism. These drugs should be better described

In line with this, the reason of an inclusion of the phentermine-topiramate combination should be given.

I'm wondering whether the Authors have information regarding the age of patients with drug-induced adverse reactions? Also were there any neurological disorders in the history. This would improve the paper by making the results more reliable

Figures should have their legends below.

moderate english langguage corrections are required

Author Response

             Chieti (IT), July 01st, 2023

REVIEWER COMMENTS:

Reviewer 1

Comments and Suggestions for Authors

Since the Authors write about incretin drugs, some additional information should be provided regarding their mechanism. These drugs should be better described

In line with this, the reason of an inclusion of the phentermine-topiramate combination should be given.

I'm wondering whether the Authors have information regarding the age of patients with drug-induced adverse reactions? Also were there any neurological disorders in the history. This would improve the paper by making the results more reliable

Figures should have their legends below.

We would like to thank the reviewer for his/her comment. Accordingly, the manuscript has been revised including a specific paragraph on incretin drugs’ mechanism of action. Regarding the comparison semaglutide and other GLP-1 receptor agonists with phentermine-topiramate, this is related to the one of the objectives of the study, which is comparing through a pharmacovigilance study semaglutide and other potentially abusable molecules currently used to promote weight loss; indeed, semaglutide, which has as its first indication as an adjunct to diet and exercise to improve glycemic control in adults with type 2 diabetes mellitus and approved in 2021 for chronic weight management in adults with obesity or overweight with at least one weight-related condition (such as high blood pressure, type 2 diabetes, or high cholesterol), has been studied in comparison with other incretin mimetics such as the following molecules, albiglutide; dulaglutide; exenatide; liraglutide; lix-isenatide; tirzepatide; and phentermine-topiramate, which has the primary indication of promoting weight loss in obese individuals. This explanation has been included in the manuscript in order to clarify the aim of the study and then discussed in the Discussion section.

Moreover, demographics have been described in the Results section and a table added. Unfortunately, no concomitant diagnosis or neurological disorders were available. Finally, figure 1 and Table have been checked and modified.

Reviewer 2 Report

This study investigated pharmacovigilance reports on adverse events for the GLP1 analogue semaglutide with a focus on reports associated with misuse/off label use, and compared those with similar reports of other GLP1 analoga and the phentermine-topiramate combination.

The topic is of interest and despite the limitations of analyses based on pharmacovigilance reports it could provide some insights regarding potential signals that might be associated with abuse and thus contribute to timely preventive measures. However, some of the findings are not presented in a very clear way and some of the reasoning and conclusions are difficult to follow based on the current version

Somer comments and suggestions:

Abstract:

-       Here, but also in the introduction and discussion sections is not very clear why you chose phentermine-topiramate as a comparator.

-       Here and throughout the manuscript you sometimes use the term “antidiabetics” although you probably mean “other GLP1 analoga” (except semaglutide). This needs to be adjusted, since “antidiabetics” is a much wider term that would have included many other substances.

Introduction:

-       Abbreviations to be explained after their first use (e.g. GIP).

-       Classical medications for obesity management can be typically divided into a few groups: opiate antagonists (naltrexone); pro-dopaminergic drugs (bupropion, phendimetrazine, benzphetamine, diethylpropion, and phentermine), fat blockers (orlistat), metformin; lorcaserine, sibutramine, and rimonabant, which was lately dismissed from the market for severe human health risks”: You can add the drug class also for the rest of the substances (sibutramine, rimonabant etc.) as well as the health risks associated with rimonabant use.

-       “…some of these molecules are classified as controlled substances [18-19]. Indeed, metformin has been associated with …”: Is metformin a controlled substance? (At the moment it reads like this but I am not sure if this is the case,)

-       It is currently not clear, here and in other parts of the manuscript, why semaglutide and not also other GLP1 analoga should be associated with misuse and why semaglutide was the focus of the study,

Results:

-       «From January 2018 to December 2022…»: What was the reason for choosing this time period and not starting earlier?

-       Figure 1: Since misuse is the main focus of the study, it would be ineteresting to present similar data for the more relevant terms (misuse etc.)

-       Table 1 & 2: You could also add the % in comparison for the total reports for each reaction (some of them are already mentioned in the text).

-       Table 3: You could also present the numbers for these, similar to Table 1. Also some explaination is needed what the IC025 represents in association with pharmacovigilance reports.

-      The FDR abbreviation is currently explained only later, at the end of the manuscript.

Discussion:

-       You found an increase of the reported AE for semaglutide compared to other substances but couldn’t a possible explanation for this also be that its sales increased during this time, leading to more pople using it and more AE being reported for this specific substance? Do you have any data on the sales of the different GLP1 RA druing this time period?

-       Also in the discussion no explanation or hypothesis is offered why semaglutide but not other GLP1 analoga should be associated with misuse.

-       Also not clear why phentermine-topiramate was chose as a comparator since it doesn’t seem that there is enough data associating it with misuse, currently described in the text as having a “potential of misuse”. Wouldn’t another substance with better described misuse risk and cases be a better comparator?

Methods:

-       A descriptive analysis of the characteristics of AE reports, including sociodemographic data, country of origin, most common diagnoses, routes of administration and concomitant licit/illicit substances was here performed”: These data are currently not shown.

-       Pharmacovigilance reporting measures, including reporting odds ratio (ROR); proportional reporting ratio (PRR); information component (IC)”: Provide some more information about how those were calculated and what they represent

Minor editing of English language required

Author Response

             Chieti (IT), July 01st, 2023

Reviewer #2:

This study investigated pharmacovigilance reports on adverse events for the GLP1 analogue semaglutide with a focus on reports associated with misuse/off label use, and compared those with similar reports of other GLP1 analoga and the phentermine-topiramate combination.

The topic is of interest and despite the limitations of analyses based on pharmacovigilance reports it could provide some insights regarding potential signals that might be associated with abuse and thus contribute to timely preventive measures. However, some of the findings are not presented in a very clear way and some of the reasoning and conclusions are difficult to follow based on the current version

Some comments and suggestions:

Abstract:

-       Here, but also in the introduction and discussion sections is not very clear why you chose phentermine-topiramate as a comparator.

-       Here and throughout the manuscript you sometimes use the term “antidiabetics” although you probably mean “other GLP1 analoga” (except semaglutide). This needs to be adjusted, since “antidiabetics” is a much wider term that would have included many other substances.

 We would like to thank the reviewer for his/her useful comments and suggestions.

 Regarding the comparison semaglutide and other GLP-1 receptor agonists with phentermine-topiramate, it is related to the objective of comparing through a pharmacovigilance study potentially abusable molecules currently used to promote weight loss; semaglutide, which has as its first indication as an adjunct to diet and exercise to improve glycemic control in adults with type 2 diabetes mellitus and approved in 2021 for chronic weight management in adults with obesity or overweight with at least one weight-related condition (such as high blood pressure, type 2 diabetes, or high cholesterol), has been studied in comparison with other incretin mimetics such as the following molecules, albiglutide; dulaglutide; exenatide; liraglutide; lix-isenatide; tirzepatide; and phentermine-topiramate, which has the primary indication of promoting weight loss in obese individuals. This explanation has been included in the manuscript in order to clarify the aim of the study and then discussed in the Discussion section.

Finally, the term “antidiabetics” has been checked and modified to “other GLP1 analoga” throughout the manuscript.

Introduction:

-       Abbreviations to be explained after their first use (e.g. GIP).

-       “Classical medications for obesity management can be typically divided into a few groups: opiate antagonists (naltrexone); pro-dopaminergic drugs (bupropion, phendimetrazine, benzphetamine, diethylpropion, and phentermine), fat blockers (orlistat), metformin; lorcaserine, sibutramine, and rimonabant, which was lately dismissed from the market for severe human health risks”: You can add the drug class also for the rest of the substances (sibutramine, rimonabant etc.) as well as the health risks associated with rimonabant use.

-       “…some of these molecules are classified as controlled substances [18-19]. Indeed, metformin has been associated with …”: Is metformin a controlled substance? (At the moment it reads like this but I am not sure if this is the case)

-       It is currently not clear, here and in other parts of the manuscript, why semaglutide and not also other GLP1 analoga should be associated with misuse and why semaglutide was the focus of the study.

            We would like to thank the reviewer for his/her comment. Accordingly, the manuscript has been revised. Specifically, abbreviations have been checked throughout the manuscript and explained after their first use. The drug class for the rest of the substances (sibutramine, rimonabant etc.) selected and the health risks associated with rimonabant use have been added. With regard to metformin, it is not categorized as a controlled substance. This information has been clarified in the manuscript. Finally, the issues on the misuse of semaglutide and the other molecules selected have been explained in the Introduction section.

Results:

-       «From January 2018 to December 2022…»: What was the reason for choosing this time period and not starting earlier?

-       Figure 1: Since misuse is the main focus of the study, it would be interesting to present similar data for the more relevant terms (misuse etc.)

-       Table 1 & 2: You could also add the % in comparison for the total reports for each reaction (some of them are already mentioned in the text).

-       Table 3: You could also present the numbers for these, similar to Table 1. Also some explaination is needed what the IC025 represents in association with pharmacovigilance reports.

-      The FDR abbreviation is currently explained only later, at the end of the manuscript.

We would like to thank the reviewer for his/her comments. In reference to the first question, as for the study period, it was selected based on the latest reported trends in drug use, e.g., semaglutide has been on the market since 2012, but only recently has been found a misuse issue related to its promotion as weight-loss treatment in non-obese people.

With regard to Figure 1, we appreciate the comment and the suggestion of the reviewer; unfortunately, we have not analyzed trends over time of the specific AE misuse; in fact, due to the small number of AER compared to the entire substance-specific database, the figure would have been unclear. Because of the importance of the issue, we preferred to pool the AERs into a single figure to make the reader aware of the increasing trend of interest and reporting of drug-related AE analyzed in the study.

Secondly, Tables have been modified according to the suggestions. Table 5 has been added and clarifications on pharmacovigilance measures inserted at the end of the manuscript.

Finally, abbreviations have been checked throughout the manuscript and explained after their first use.

Discussion:

-       You found an increase of the reported AE for semaglutide compared to other substances but couldn’t a possible explanation for this also be that its sales increased during this time, leading to more people using it and more AE being reported for this specific substance? Do you have any data on the sales of the different GLP1 RA druing this time period?

-       Also in the discussion no explanation or hypothesis is offered why semaglutide but not other GLP1 analoga should be associated with misuse.

-       Also not clear why phentermine-topiramate was chose as a comparator since it doesn’t seem that there is enough data associating it with misuse, currently described in the text as having a “potential of misuse”. Wouldn’t another substance with better described misuse risk and cases be a better comparator?

 We would like to thank the reviewer for his/her comments. The points raised are interesting and have been clarified throughout the manuscript (Introduction and Discussion sections). First of all, although we do not have access to prescription/sales worldwide data, which are normally restricted to institutions and subject to fees, from the information available online an increase in the sale of GLP-1-RA has been reported starting from 2016, especially for liraglutide, dulaglutide, and semaglutide; and this data is expected to grow in the next years; accordingly, the manuscript has been modified as follows: “Consistent with this, in the years previous to the outlined timeframe (i.e. 2018-2022) considered here, prescriptions for all the molecules selected increased, but this was especially true for those relating to liraglutide, dulaglutide, and semaglutide”.

With regard to the second point, we may hypothesize pharmacological differences, in terms of pharmacokinetic and pharmacodynamics properties, may have an influence on the misuse issue of the molecules involved in the study. We have commented on this in the Discussion section as follows “The information provided by this study does not allow us to deduce explanations for the misuse of each individual substance, but, on the basis of pharmacovigilance signals, to compare molecules with each other and provide useful information for clinicians and institutions to monitor possible side effects, adverse events and misuse. We can hypothesise that explanations related to the formulation (subcutaneous versus oral), availability and prescribability, and pharmacokinetic and pharmacodynamic properties, as well as the effects on weight reduction (semaglutide and liraglutide would appear to be the most effective in the long term) of the individual drugs may explain the misuse of semaglutide in comparison with the other molecules under study”.

Finally, regarding the comparison semaglutide and other GLP-1 receptor agonists with phentermine-topiramate, it is related to the objective of comparing through a pharmacovigilance study potentially abusable molecules currently used to promote weight loss; semaglutide, which has as its first indication as an adjunct to diet and exercise to improve glycemic control in adults with type 2 diabetes mellitus and approved in 2021 for chronic weight management in adults with obesity or overweight with at least one weight-related condition (such as high blood pressure, type 2 diabetes, or high cholesterol), has been studied in comparison with other incretin mimetics such as the following molecules, albiglutide; dulaglutide; exenatide; liraglutide; lix-isenatide; tirzepatide; and phentermine-topiramate, which has the primary indication of promoting weight loss in obese individuals. This explanation has been included in the manuscript in order to clarify the aim of the study and then discussed in the Discussion section.

Methods:

-       “A descriptive analysis of the characteristics of AE reports, including sociodemographic data, country of origin, most common diagnoses, routes of administration and concomitant licit/illicit substances was here performed”: These data are currently not shown.

-       “Pharmacovigilance reporting measures, including reporting odds ratio (ROR); proportional reporting ratio (PRR); information component (IC)”: Provide some more information about how those were calculated and what they represent

We would like to thank the reviewer for his/her comment. Accordingly, the manuscript has been revised. A specific paragraph on pharmacovigilance measures has been added in the Methods section.

Round 2

Reviewer 1 Report

the manuscript has been improved, therefore, in my opinion, it is suitable for being published

Author Response

Thank you, reviewer, for your comment.

Reviewer 2 Report

Thank you for addressing the majority of the comments. Some points that were not or not adequately addressed in the context of the revision:  

-       The term “antidiabetics” is still used instead of “other GLP1 analoga” in some sections, e.g. title of Table 3.

-       Tables 1-3: Are the currently shown % based on the comparison of the total number of AER overall or for each substance/substance group? To be clarified.

Table 4: What do the numbers in parentheses represent? To be clarified. IC025 totally not explained in the footnote.

Minor editing required.

Author Response

Chieti (IT), July 04th, 2023

To:  

Ms. Wesley Zhao

Assistant Editor

E-Mail: [email protected]

RE:  Is there a risk for Semaglutide Misuse? Focus on the Food and Drug Administration-FDA Adverse Events Reporting System (FAERS) pharmacovigilance dataset (pharmaceuticals-2467025)

Dear Editor,

Thank you for allowing us to submit a revised version of the manuscript entitled " Is there a risk for Semaglutide Misuse? Focus on the Food and Drug Administration-FDA Adverse Events Reporting System (FAERS) pharmacovigilance dataset "being considered for Pharmaceuticals.

We have revised the manuscript whilst carefully considering each item the Reviewers raised in detail. The manuscript has undoubtedly benefitted from the insightful suggestions provided.

All changes have been tracked by underlining them throughout the manuscript. Please find below our detailed answers to each of the advice/suggestions provided.

Looking forward to hearing from your earliest convenience,

Yours sincerely,

Stefania Chiappini, MD, PhD

Department of Neuroscience, Imaging and Clinical Sciences

University “G. d’Annunzio”, Chieti.

[email protected]

REVIEWER COMMENTS:

Reviewer 3

Thank you for addressing the majority of the comments. Some points that were not or not adequately addressed in the context of the revision: 

 -The term “antidiabetics” is still used instead of “other GLP1 analoga” in some sections, e.g. title of Table 3.

-Tables 1-3: Are the currently shown % based on the comparison of the total number of AER overall or for each substance/substance group? To be clarified.

-Table 4: What do the numbers in parentheses represent? To be clarified. IC025 totally not explained in the footnote.

Comments on the Quality of English Language

Minor editing required.

We would like to thank the reviewer for his/her comments and suggestions. Accordingly, we have modified the term “antidiabetics” to “other GLP-1 analogues” in Tables 3 and 4. With regard to Tables 1-3 % shown are based on the comparison of each substance/substance group. Referring to Table 4, as described, numbers in parentheses represent the False Discovery Rate (FDR), and boldface denotes significance at FDR<0.05. The IC025 is now explained in the footnote.

Finally, the English has been checked troughout the manuscript.